# A New Acoustical Autonomous Method for Identifying Endangered Whale Calls: A Case Study of Blue Whale and Fin Whale

**DOI:** 10.3390/s23063048

**Published:** 2023-03-12

**Authors:** Farook Sattar

**Affiliations:** Department of Electrical and Computer Engineering, University of Victoria, Victoria, BC V8W 2Y2, Canada; fsattar@ieee.org

**Keywords:** whale calls, marine bioacoustics, endangered whale, deep learning, artificial intelligence, wavelet scattering transform, identification, small data set

## Abstract

In this paper, we study to improve acoustical methods to identify endangered whale calls with emphasis on the blue whale (*Balaenoptera musculus*) and fin whale (*Balaenoptera physalus*). A promising method using wavelet scattering transform and deep learning is proposed here to detect/classify the whale calls quite precisely in the increasingly noisy ocean with a small dataset. The performances shown in terms of classification accuracy (>97%) demonstrate the efficiency of the proposed method which outperforms the relevant state-of-the-art methods. In this way, passive acoustic technology can be enhanced to monitor endangered whale calls. Efficient tracking of their numbers, migration paths and habitat become vital to whale conservation by lowering the number of preventable injuries and deaths while making progress in their recovery.

## 1. Introduction

It is of utmost importance to conserve endangered whale populations which are declining due to various reasons, such as striking by ships/vessels, entangling with fishing gear, and global warming. Automated acoustic monitoring has been used for monitoring marine species such as whales. Recorded acoustic samples let us listen and analyze sounds from soniferous whales for their identification. As sound travels quicker than light underwater, it is good to do acoustic monitoring than video surveillance for identifying endangered whales. However, accurate acoustic identification of endangered whale calls (vocalizations) is still difficult, especially when a whale population is getting dangerously small and the size of the available data samples is also too small.

Blue whales (*Balaenoptera musculus*) are the largest of the baleen whales and are endangered worldwide [1]. Blue whale calls are low-frequency (20–100 Hz) and repetitive [2]. Blue whales are known to produce downswept FM (frequency-modulated) calls that are often referred as D-calls. Both male and female blue whales have been found to produce such calls [3]. On the contrary, it is observed that only males produce song, and with that, these calls are associated with the breeding season. Thus, blue whale song apparently carries information about the population. Similarly, fin whales (*Balaenoptera physalus*) are listed as endangered species, which also produce low-frequency vocalizations (i.e., <100 Hz) [4,5]. Single vocalizations, in particular, are generated by male fin whales, whereas songs in the form of pulse trains can occur at high sound pressure that can be detected over a long distance (e.g., >50 km) [6]. In locations of high fin whale density, the songs and single vocalizations of numerous fin whales do overlap in time and frequency, producing the so-called fin whale chorus [6].

The development of robust deep learning methods to identify whales or finding when and where each whale population occurs is getting much attention. Recent abundance estimates using acoustic whale calls can aid assessment of the current status of each identified whale population, especially for blue whales and fin whales whose population sizes are critically decaying. This status assessment thus provides us the basis to a proper management plan for the conservation of endangered whale populations.

Deep learning motivation is greatly deduced by artificial intelligence (AI), which simulates the ability of the human brain in terms of analyzing, making decisions, and learning. The AI-enabled technique, such as deep learning, is quickly becoming a mainstream technology that can analyze large volumes of data and potentially unlock insights, especially in ocean monitoring applications. Deep learning can be defined as a technique of machine learning to learn useful features directly from given sounds, images, and texts. The core of deep learning is hieratically computing features and representing information, such as defining the features starting from a low level to high level. Different hidden layers are involved in making decisions by using the feedback from one layer to the previous one, or the resulting layer will have been fed into the first layer. Therefore, many layers are exploited by deep learning for nonlinear data processing of supervised or unsupervised feature extraction for classification and pattern recognition. It is difficult for a computer to understand complex data, such as an image or a sequence of data of a complex nature, so deep learning algorithms are used instead of usual learning methods. The conventional methods have been overtaken by deep learning methods which can detect and classify objects in complex scenarios. Deep learning can thus help us to create better ocean acoustic detection and classification models.

Recently, a few studies have been reported about the deep learning methods developed for blue whale and fin whale monitoring. In the Ref. [2], Siamese neural networks (SNN) were utilized to detect/classify blue whale calls from the acoustic recordings. The method classified calls from four populations of blue whales providing the highest accuracy of 94.30% for Antarctic blue whale calls, while the lowest accuracy of 90.80% was provided for the central Indian Ocean (CIO) blue calls. Studies in the Ref. [3] showed that the DenseNet-automated blue whale D-Call detector, which is based on conventional Convolutional Neural Networks (CNN) provided better results in terms of detection probability than that of human observers’ analyses, particularly at low and medium signal-to-noise ratios (SNRs). Higher detection probabilities (0.905 and 0.910 for low and medium SNRs) were obtained compared to the detection probabilities obtained by human observers’ analyses (0.699 and 0.697 for low and medium SNRs). In the Ref. [3], a long-term recording dataset, particularly the Australian Antarctic Division’s “Casey2019” dataset was used for the results. In the Ref. [7], a two-stage deep-learning approach was developed based on a region-based convolutional neural network (rCNN) and following CNN to automatically detect/classify both blue whale D-calls and fin whale 40-Hz calls. In stage 1, the detection of regions of interests (ROIs) containing potential calls was performed using rCNN. In stage 2, a CNN was employed to classify the target whale calls from the detected ROIs. The work in the Ref. [8] presents an application of deep learning for automatic identification of fin whale 20 Hz calls, which are sometimes contaminated with other sources, such as shipping and earthquakes. All of these recently proposed advanced deep-learning-based methods have one common feature—they all require a large dataset for learning.

In this paper, taking advantage of deep nonlinear features of wavelet scattering transform (WST) [9], we adopted the LSTM [10] deep learning classifier to automatically detect/classify the endangered whale calls. The proposed method was evaluated for real recorded ocean acoustic data from the northeast (NE) Pacific, giving high performances in terms of classification results with a small dataset.

The main contributions of this work are the following: (1) Study of the applicability of the WST to detect/classify endangered whale vocalizations with a small dataset. (2) Incorporation of the temporal contextual information provided by the LSTM network for identification of endangered whale calls. (3) Proposal of an efficient deep-learning-based endangered whale monitoring method from small data samples. (4) To the best of our knowledge, the WST and LSTM techniques together have not been explored for identification of endangered whale calls.

The organization of the paper is as follows. Section 2 presents the materials and methods related to this study. In Section 3, the experimental results are described. The discussion is provided in Section 4.

## 2. Materials and Methods

### 2.1. Dataset

The dataset was constructed using a csv file containing more than 8000 manual annotations for marine mammals (e.g., various whales). Each annotation shows the timestamp (i.e., the start time and the end time), class type and the name of the original wav file from Ocean Networks Canada (ONC)’s database [11]. Each annotated wav file has a duration of 5 min for a sampling frequency of 64 kHz. Prior to doing the annotation, the original files were segmented and cropped into clips. While the same file name can be referred to many of the annotations within the ONC database, the corresponding cropped files were named using a hash of the original csv file’s annotation properties. The hashed file name provides the following advantages. First, it gives a unique filename to each annotation. Second, it restricts the processed scripts for downloading and cropping the annotations.

For manual annotation, spectrogram analysis was carried out for recordings every 5 min in Audacity version 2.0.6 [12] software when different whale calls were labeled by an expert from ONC. In total, our generated dataset belonged to 8728 annotated files where files associated with a single class label were considered for use here. In addition, only endangered baleen whales of blue whale and fin whale were taken into account in this study. These data constraints made our dataset reduce to 932 files containing single labels, and the distribution of those files is shown in Table 1. These recordings contain various activities from which we have considered here the activities of the blue whale for 20–100 Hz [2] and the fin whale for 5–100 Hz [5,13].

### 2.2. Data Analysis

The following stages are taken into account for detection/classification of the endangered whale calls.

#### 2.2.1. Data Preprocessing

Firstly, we reduced the sampling rate of the data segments from 64,000 Hz to 6400 Hz. Secondly, we resized the resampled data segments into data blocks with lengths of *N* samples, which is set here as *N* = 64,000 (10 s). To reduce the end effect, each resized block was time-windowed using a *N*-sample Hamming window [14]. It should be noted that computational complexity can be reduced through resampling due to the processing of data at a lower sampling rate. On the other hand, resizing leads to saved memory by compressing the signal along time without modifying its spectral content [15]. Figure 1 shows a flow diagram of the signal processing steps employed in the preprocessing stage. Here, the resampling was performed using a polyphase anti-aliasing filter, whereas the resizing was performed by a “nearest-neighborhood” interpolation.

#### 2.2.2. Feature Extraction

The wavelet scattering transform (WST) coefficients are utilized here for feature extraction [16,17]. The 1D WST is computed by cascading wavelet transforms along with nonlinear complex modulus operations followed by average filtering. The WST of a 1D signal z(t) can be represented as
(1)$$S_{J}z=\left[S_{J}^{\left(0\right)}z,\;\;S_{J}^{\left(1\right)}z,\;\;S_{J}^{\left(2\right)}z\right]$$
where
$$S_{J}^{\left(0\right)}z\left(t\right)=z*\phi_{J},$$
$$S_{J}^{\left(1\right)}z\left(t,\lambda \right)=\left|z*\psi_{\lambda }^{\left(1\right)}\right|*\phi_{J},\;\;and$$
$$S_{J}^{\left(2\right)}z\left(t,\lambda ,\mu \right)=||z\;*\;\psi_{\lambda }^{\left(1\right)}\left|\;*\;\psi_{\mu }^{\left(2\right)}\right|*\phi_{J}$$

In Equation (Equation 1), ‘∗’ denotes the convolution operator, $\psi_{\lambda }^{\left(1\right)}$ and $\psi_{\mu }^{\left(2\right)}$ are the filters representing complex wavelets having center frequencies λ and μ, whereas $\phi_{J}\left(t\right)$ is a real lowpass filter with zero-mean frequency.

The implementation of the 1D scattering transform is performed for a given set of wavelet filters whose parameter values are specified initially. Hence, the wavelets are fixed, however, there may be changes for the other parameters after the goal is set, for instance, whether all of $S_{J}^{\left(0\right)}z$, $S_{J}^{\left(1\right)}z$, and $S_{J}^{\left(2\right)}z$, or just $S_{J}^{\left(0\right)}z$ and $S_{J}^{\left(1\right)}z$ would be computed.

While a given input signal length is *N*, the maximum scale of the WST is set to $2^{J}$. The other issues are the time-frequency resolutions of the wavelets. It is set to *Q* = 8 wavelets per octave for the first-order wavelets, $\psi_{\lambda }^{\left(1\right)}$. On the other hand, it is *Q* = 1, that is, one wavelet per octave as always for the second-order wavelets $\psi_{\mu }^{\left(2\right)}$. These configurations are set to preserve the most signal information for classification.

It is worth noting that postprocessing was performed for the scattering coefficients. Therefore, log-scattering coefficients were obtained by taking the logarithm values for the scattering vectors of the first-order and the second-order wavelets. This can facilitate building the models with reduced dynamic range and stable variability. Moreover, log-scattering coefficients can be well-suited for audio classification due to the fact that amplitudes of the audio signals then vary across several orders of magnitude with no significant changes in the signal content. This process can be characterized through use of the Weber–Fechner law [18] in psychoacoustics.

#### 2.2.3. Classification Method

The classification was performed using Long Short-Term Memory (LSTM) [19,20]. It is a recurrent neural network (RNN) containing an input gate, forget gate, output gate, temporal forward pass and backpropagation. The input gate, forget gate and output gate responses at time *t* denoted by $i^{t}$, $o^{t}$, and $f^{t}$, respectively, are associated with the forward pass in a LSTM architecture and can be expressed as:(2)$$i^{t}=\mathrm{Sigmoid}\left(W_{ih}h^{\left(t-1\right)}+W_{ix}x^{t}+b_{i}\right)$$
(3)$$o^{t}=\mathrm{Sigmoid}\left(W_{oh}h^{\left(t-1\right)}+W_{ox}x^{t}+b_{o}\right)$$
(4)$$f^{t}=\mathrm{Sigmoid}\left(W_{fh}h^{\left(t-1\right)}+W_{fx}x^{t}+b_{f}\right)$$

In Equations (Equation 2)–(Equation 4), $h^{\left(t-1\right)}$ refers to the hidden state at time (t−1), $W_{ih}$, $W_{oh}$, and $W_{fh}$ are the weights associated with $h^{\left(t-1\right)}$ for the corresponding gates, $b_{i}$, $b_{o}$, and $b_{f}$ are the respective bias vectors, $Sigmoid\left(x\right)=\frac{1}{1+e^{\left(1-x\right)}}$ is the activation function.

The following formulations are also associated with the forward pass:(5)$$d^{t}=\mathrm{Tanh}\left(W_{dh}h^{\left(t-1\right)}+W_{dx}x^{t}+b_{d}\right)$$
(6)$$c^{t}=f^{t}⊙c^{\left(t-1\right)}+i^{t}⊙d^{t}$$
(7)$$h^{t}=o^{t}⊙Tanh\left(c^{t}\right)$$
(8)$$L^{t}=\phi \left(h^{t}\right)$$
(9)$$L=\sum_{t=1}^{T}L^{t}$$
where $d^{t}$ stands for the distorted input to the memory cell at time *t*, $W_{dh}$ is the weight associated with $h^{\left(t-1\right)}$ and $b_{d}$ is the corresponding bias vector, Tanh(·) is the activation function, $c^{t}$ refers to the state of the memory cell at time *t*, $h^{t}$ denotes the hidden state at time *t*, and ‘⊙’ stands for point-wise multiplication. Additionally, in Equation (Equation 8), ϕ maps the hidden state to the network loss $L^{t}$ at time *t*. Then the total network loss *L* is found by adding each individual network loss $L^{t}$ along time, as depicted in Equation (Equation 9).

In order to optimize the LSTM model, backpropagation through time was implemented and the most critical value to calculate in LSTM is:(10)$$\frac{\partial L}{\partial c^{t}}=\sum_{t=1}^{T}\frac{\partial L^{t}}{\partial c^{t}}$$

A critical iterative property was adopted to calculate the above value:(11)$$\frac{\partial L}{\partial c^{\left(t-1\right)}}=\frac{\partial L}{\partial c^{t}}\frac{\partial c^{t}}{\partial c^{\left(t-1\right)}}+\frac{\partial L^{\left(t-1\right)}}{\partial c^{\left(t-1\right)}}$$

Several other LSTM gradients can be calculated through the chain rule using the above calculation output:(12)$$\frac{\partial L}{\partial o^{t}}=\frac{\partial L}{\partial h^{t}}\frac{\partial h^{t}}{\partial o^{t}},$$
(13)$$\frac{\partial L}{\partial i^{t}}=\frac{\partial L}{\partial c^{t}}\frac{\partial c^{t}}{\partial i^{t}},$$
(14)$$\frac{\partial L}{\partial f^{t}}=\frac{\partial L}{\partial c^{t}}\frac{\partial c^{t}}{\partial f^{t}},$$
(15)$$\frac{\partial L}{\partial d^{t}}=\frac{\partial L}{\partial c^{t}}\frac{\partial c^{t}}{\partial d^{t}}$$ (see [21] for more details).

## 3. Results

The experimental setup and the corresponding detection/classification results of the proposed method, as well as relevant state-of-the-art methods, are presented in the following.

### 3.1. Training Data and Test Data

The dataset used for our results consists of 217 hydrophone recordings for blue whale calls and 715 hydrophone recordings for fin whale calls (sampled at 64,000 Hz) (see Table 1). This dataset includes the noisy signals containing blue/fin whale calls. In order to obtain noise-only recordings, we performed zero-phase filtering for each of the recordings using a fourth-order highpass Butterworth filter with a cut-off frequency of 2000 Hz. Therefore, we have the same number of noise-only recordings. These recordings are then concatenated when three combined datasets containing recordings with and without blue whale calls, with and without fin whale calls, with blue whale and with fin whale calls, are used for our simulations.

For training and testing purposes, each feature set was partitioned into two subsets, namely, the training feature set and test feature set. We used 50% of the data for training and the other 50% for testing in all our simulations. Then the feature sets were standardized with zero-mean and unit variance before input into the classifier. The results were obtained in terms of mean classification results over 100 different trials. For each trial we used different training and test datasets whose configurations changed randomly.

### 3.2. Analytic Result

The spectrograms of the three noisy recordings containing blue whale calls are shown in Figure 2a–c, while the corresponding noise-only signals are presented in Figure 2d–f. Each spectrogram was configured for an input signal of 10 s by using a Hann window of a length of 1600 samples (250 ms) with 75% overlap when the sampling frequency of the signal was downsampled to 6400 Hz.

Similarly, the spectrograms of the three noisy recordings containing fin whale calls are depicted in Figure 3a–c, whereas the corresponding noise-only signals are shown in Figure 3d–f. Each spectrogram was plotted for an input signal of 10 s by using a Hann window of length 1600 samples (250 ms) with 75% overlap, whereas the sampling frequency of the signal was downsampled to 6400 Hz, the same as in Figure 2.

We used a three-layer WST and chose Morlet (Gabor) wavelets [22], a commonly used complex wavelets due to its simple mathematical representation and good localization. The framework has two filter banks when the number of layers is three. The quality factors (i.e., the number of wavelet filters per octave) for the first and the second filter banks were set to *Q* = 8 and 1, respectively.

For an input signal of length *N* = 64,000 samples and the *Q* values as above, the output of the framework is a feature matrix with size (246 × 8 ×2). The feature matrix is then formed with 246 scattering paths and 8 scattering time windows for both the real and imaginary parts of the signal. Hence, the feature set contains 492 feature vectors with dimension 8, while we have excluded the feature vectors associated with path 1. For an *M* number of signals, a three-dimensional feature output of size (492 × 8 × M) was thereby generated. In order to build the feature set for the classifier, we multiplied the values of 492 and 8 to reduce them to a 1D sequence and thereby convert the feature output for *M* signals from three dimensions to two dimensions.

In our training process, we have chosen the following parameters for the LSTM classifier: the number of hidden layers = 512, learning rate = 0.0001, minibatch size = 128, and Adam (Adaptive Moment Estimation) optimizer to train the model. Note that the above parameter setting for the LSTM network provides us good results and the whole process is implemented in MATLAB2022b [23].

The performances of the method are evaluated in terms of classification accuracy (%) as well as sensitivity (%) and specificity (%) as shown below:(16)$$\mathrm{Accuracy}=\frac{\mathrm{TP}+\mathrm{TN}}{(\mathrm{TP}+\mathrm{FP})+\left(\mathrm{TN}+\mathrm{FN}\right)}$$
(17)$$\mathrm{Sensitivity}=\frac{\mathrm{TP}}{\mathrm{TP}+\mathrm{FN}}$$
(18)$$\mathrm{Specificity}=\frac{\mathrm{TN}}{\mathrm{TN}+\mathrm{FP}}$$

The average accuracy for the blue whale calls (%) over 100 trials is found to be as high as 91.06% using a single epoch in the learning process. The confusion matrix for a trial with accuracy 97.69(%) is shown in Table 2. The rows of this confusion matrix denote the true class labels and the columns represent labels for the predicted class. In the confusion matrix, the diagonal elements refer to the number of correctly classified samples for different class labels, as indicated by the corresponding row/column label. All the non-diagonal elements of the confusion matrix stand for wrongly classified classes, as in Table 2.

The average accuracy for the fin whale calls (%) over 100 trials is found to be as high as 100% for a single epoch. The confusion matrix for a trial with accuracy 100(%) is shown in Table 3.

The average accuracy for the classification of blue whale calls and fin whale calls (%) from 100 trials is achieved as 98.40% for a single epoch. The confusion matrix for a trial with accuracy 97.42(%) is displayed in Table 4.

### 3.3. The Choice of Invariance Scale

We have considered different invariance scale which is determined as the time support of the lowpass filter $\phi_{J}\left(t\right)$. Figure 4 shows the classification accuracies at different invariance scales of the WST, while we set the invariance scale of 6 (s) as providing the highest classification accuracy (%). The results illustrated in Figure 4 were obtained for the “Blue whale (BW) + Noise” dataset. As we see, the classification performances show some differences with the changes of invariance scale. When the scale is too large, the convolution partly loses the high-frequency information which could cause deterioration of the accuracy at higher scales. On the other hand, when the scale gets small, the convolution removes less noise that might cause a decrease in the performance at lower scales.

### 3.4. Results with SVM Classifier

We have further obtained the results with the support vector machine (SVM) classifier [24] and WST-based features. The SVM classifier basically finds the optimal hyperplane by processing the data using kernels. The optimal hyperplane is produced in terms of the best data separation by maximizing the margin between the decision boundary and the closed data points.

For a given dataset $S=\left\{\left(x_{i},y_{i}\right)\right\}|x_{i}\in R^{n},y_{i}\in {\left\{-1,1\right\}}_{i=1}^{m}$, where *n* refers to the dimensionality of the input data and *m* is the total number of samples, consider how the expression of a hyperplane is w·ϕ(x)=−b, where *w* is the trainable weight vector of the SVM classifier, *x* is the feature vector, ϕ(·) is the kernel function, and *b* is a bias. Thus, if the point (x,y) is on the hyperplane, w·ϕ(x)+b=0; if the point (x,y) is not on the hyperplane, the value of w·ϕ(x)+b could be either >0 (positive) or <0 (negative) (considering two classes for binary classification problem optimization). The two SVM parameters, that is, the regularization parameter C(>0) and the kernel’s scale parameter γ are set to 10 and 0.05, respectively, where the radial basis function (RBF) is used for the kernel function. The SVM used in our simulations was the least-square SVM (LS-SVM). Additionally, the feature vectors were normalized to zero-mean and unit variance before being fed into the SVM classifier. In Table 5, the average accuracies (%) from 100 trails are presented for the three datasets.

### 3.5. Comparison Result

#### 3.5.1. Comparison I

We have compared the results with a relevant state-of-the-art method based on a short-term Fourier transform (STFT) and LSTM network [25]. The instantaneous frequency $f_{i}\left(t\right)$ was calculated from the STFT [26] using a 256-sample rectangular window (moving along time with step-size of 1) for feature extraction and used as input to the LSTM classifier. We used an in-built MATLAB function instfreq(·), which estimates the instantaneous frequency to be the first-order spectral moment of the spectrogram, that is, ${\left|\mathrm{STFT}\right|}^{2}$ of an input signal (see Equation (Equation 19)).
(19)$$f_{i}\left(t\right)=\frac{\int_{0}^{\infty }f{\left|\mathrm{STFT}\left(t,f\right)\right|}^{2}df}{\int_{0}^{\infty }{\left|\mathrm{STFT}\left(t,f\right)\right|}^{2}df}\;\;\left(f:\mathrm{frequency},\;\;t:\mathrm{time}\right)$$

In Equation (Equation 19), $f_{i}\left(t\right)$ is the instantaneous frequency. The corresponding discrete form of Equation (Equation 19) is $f_{i}\left(n\right)=\sum_{0}^{K-1}k|\mathrm{STFT}{\left(n,k)\right|}^{2}/\sum_{0}^{K-1}{\left|\mathrm{STFT}(n,k)\right|}^{2}$. Here the discrete form of STFT is $\mathrm{STFT}\left(n,k\right)=\sum_{p=-P/2}^{P/2}x\left(p\right)g\left(n-p\right)e^{-j2\pi kn}$ where *x* is the signal of length *N*, *g* is the window of size (2P+1)×1, n(1≤n≤N) and k(0≤k≤K−1) refer to discrete time and frequency.

The following parameters were used for the LSTM classifier: the number of hidden layers = 100, learning rate = 0.01, minibatch size = 128, and Adam optimizer were used for training the model. Here we have considered the common choices for the values of the LSTM parameters [27].

The average accuracy of the comparison method through 100 trials was found as 69.31% for the blue whale calls. In Table 6, the corresponding confusion matrix of a certain trial is shown where the classification accuracy was obtained as 72.8% with 10 epochs in the learning process.

Similarly, the mean classification accuracy of this comparison method for the fin whale calls is found as 80.28%. Table 7 presents a confusion matrix associated with the fin whale calls for a particular trial giving 80.4% classification accuracy.

In addition, we have calculated the mean accuracy for the blue whale and fin whale calls by this comparison method, giving a result of 78.75% for 100 trials. The corresponding confusion matrix for a single trial providing a classification accuracy of 81.1% is displayed in Table 8.

#### 3.5.2. Comparison II

Here, we compared with another relevant state-of-the-art method that is based on scattergram and deep CNN (Convolutional Neural Network) [28]. The scattergram of size (n×m) was computed using WST and is similar to the mel-spectrogram when considering the filter bank 1 or layer 1 to compute the WST for finding the scattergram. On the other hand, CNN is a popular deep learning approach for classification. The CNN consists of three convolution blocks and one fully connected (FC) layer. Each convolution block is composed of a 1D convolution layer of length 3 and batch normalization. Each convolutional layer is followed by a max pooling layer, with pooling size (1×2) and stride (1, 2). The network has 8, 16 and 32 filters, respectively. A fully connected layer with *C* hidden neurons, where *C* is the number of classes to be identified, is connected to a categorical softmax layer. We used a rectified linear unit (ReLU) as the activation function in all layers. This architecture takes the scattergram of the fixed-length acoustic data being an input image. The flowchart of the CNN architecture, together with the number of filters and the size of the kernels, are presented in Figure 5.

The CNN was trained through a stochastic gradient descent (SGD) optimizer. Similar to the proposed method, a learning rate of 0.0001 and a single epoch were used for the results.

In Table 9, the average classification accuracies (%) for 100 trails and three datasets are presented as obtained by the above comparison method II.

## 4. Discussion

Through the LSTM, the temporal context inside the feature set are fully considered and the nonlinear mapping relationship between the past and future information of the signal are learned. The new approach demonstrates significant improvements in the endangered whale calls identification with high classification results as above 90% with small data samples. The method is also computationally efficient, since only a single epoch for the deep learning process is able to produce high classification accuracies (%), as shown in Table 2, Table 3 and Table 4.

The performances obtained with the fin whales were found to be slightly better than that of the blue whales in terms of higher classification scores. This could be clarified from the spectrogram plots of the illustrative blue whale and fin whale calls. For instance, the fin whale 20 Hz pulses are quite prominent in the spectrograms (see Figure 3b,c), while the blue whale B-calls (40–50 Hz) are less visible in the corresponding spectrogram plots (see Figure 2a,b). In Table 2 and Table 3, the sensitivity represents the correctly classified of the respective whale calls, and the specificity represents the correctly classified of the noise. Then the false positive rates (FPRs) obtained from the specificity (%) were found to be as low as 4.39% and 0%, respectively.

In our proposed framework, the variability is linearized by the WST providing invariants to translations through such average pooling. Most importantly, WST comprising the LSTM network can produce good identification results with small sets of training data. In fact, WST can assist us to extract significant features for LSTM in those situations when it is not possible to learn efficient features with the available training data in case of data scarcity. This makes our proposed scheme for use, such as few-shot learning [29] for identification of endangered whale calls unlike the existing recent deep-learning-based methods [2,3,7,8] that require large datasets for learning.

Both LSTM and CNN are deep neural networks, although the design mechanism of LSTM is different than a CNN. Usually, the LSTM is designed to process and perform prediction/classification from sequences of data by exploiting temporal correlation, whereas the CNN is designed to process image data (for example, a 2D scattergram image shown in Figure 5) for classification by exploiting spatial correlation. The LSTM architecture solved the zero-gradient and exploded gradient, as well as short-term memory problems in RNN. On the other hand, CNN automatically generates complex features at different layers. Basically, WST can be realized as a CNN with fixed filters. In this way, we have a new mechanism here for the classification of endangered whale calls by combining CNN and LSTM. This proposed combined scheme can be efficient, both in terms of classification accuracy and computation.

The proposed method outperforms the relevant state-of-the-art methods based on STFT and LSTM. The presented method provides much-improved classification results even with 1 epoch instead of 10 epochs as used by the comparison method in the learning process. Then the performances of the proposed method is better than the comparison method in terms of classification accuracy as well as computational burden. We have further compared our method with another relevant method based on scattergram and deep CNN, while our method yields better performance. Moreover, the proposed scheme demonstrates high noise resiliency when we compare the results of the proposed scheme and the SVM classifier-based scheme summarized in Table 5.

To the best of our knowledge, the framework consisting of the WST and LSTM networks has not been investigated for detection/classification of whale calls. The preliminary results are presented in this paper. For future work, we would like to obtain data from noisier environments to make the proposed method more robust by proposing a learned wavelet scattering transform together with optimizing the model parameters of the LSTM network. Moreover, we plan to investigate the method for other critically endangered species, such as the North Atlantic right whale and Sei whale.

Finally, we want to emphasize that the MATLAB source codes will be available upon request to the corresponding author.

## Figures and Tables

**Figure 1 sensors-23-03048-f001:**
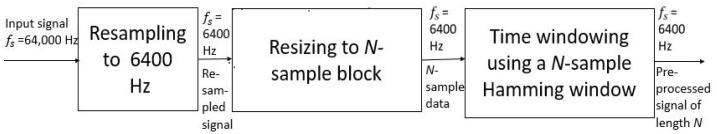
Block diagram of the preprocessing steps.

**Figure 2 sensors-23-03048-f002:**
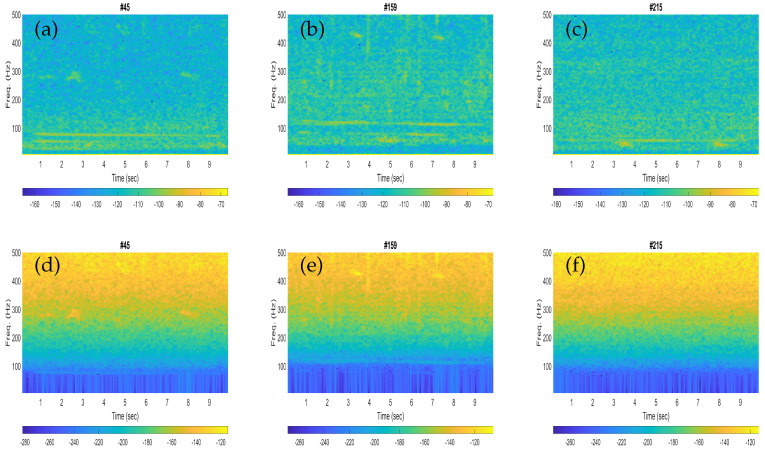
Illustrative spectrogram plots for the noisy signals with blue whale calls (**a**–**c**) and the corresponding noise-only signals (**d**–**f**).

**Figure 3 sensors-23-03048-f003:**
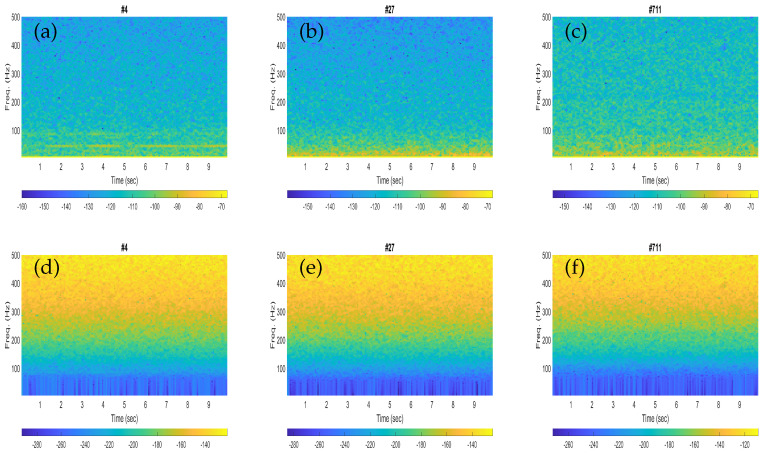
Illustrative spectrogram plots for the noisy signals with fin whale calls (**a**–**c**) and the corresponding noise-only signals (**d**–**f**).

**Figure 4 sensors-23-03048-f004:**
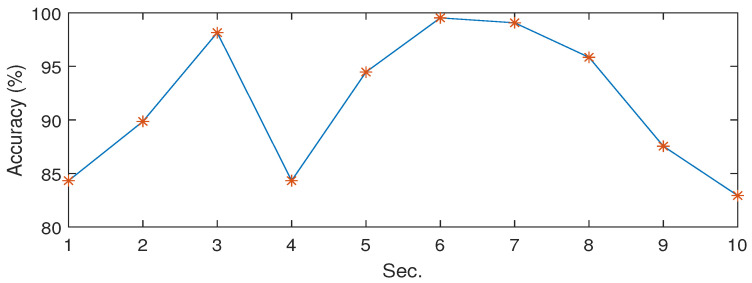
The results of classification for various invariance scales.

**Figure 5 sensors-23-03048-f005:**
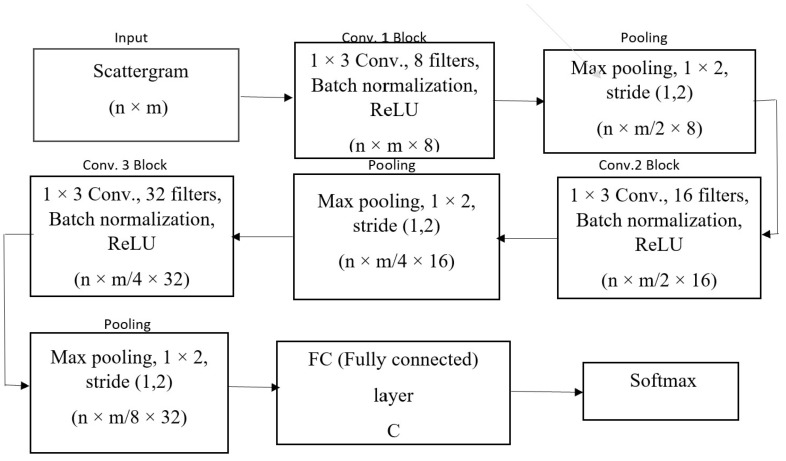
The CNN architecture used.

**Table 1 sensors-23-03048-t001:** The number of sound (wav) files used.

Endangered Whale	No. of Recordings
Blue whale (BW)	217
Fin whale (FW)	715

**Table 2 sensors-23-03048-t002:** The confusion matrix for the proposed method with the ‘Blue whale (BW) + Noise’ dataset (the accuracy (%) is indicated in bold font and calculated as a ratio of the sum of diagonal values to the sum of all values × 100).

Predicted class				
True class		Blue Whale (BW)	Noise	Sensitivity (%)
	Blue whale (BW)	103	5	95.37
	Noise	0	109	100
	Specificity (%)	100	95.61	97.69

**Table 3 sensors-23-03048-t003:** The confusion matrix for the proposed method with the ‘Fin whale (FW) + Noise’ dataset (the accuracy (%) is indicated in bold font and calculated as a ratio of the sum of diagonal values to the sum of all values × 100).

Predicted class				
True class		Fin Whale (FW)	Noise	Sensitivity (%)
	Fin whale (FW)	354	0	100
	Noise	0	361	100
	Specificity (%)	100	100	100

**Table 4 sensors-23-03048-t004:** The confusion matrix for the proposed method with the ‘Blue whale (BW) + Fin whale (FW)’ dataset (the accuracy (%) is indicated in bold font and calculated as a ratio of the sum of diagonal values to the sum of all values × 100).

Predicted class				
True class		Blue Whale (BW)	Fin Whale (FW)	Sensitivity (%)
	Blue whale (BW)	112	11	91.06
	Fin whale (FW)	1	342	99.71
	Specificity (%)	99.12	96.88	97.42

**Table 5 sensors-23-03048-t005:** The average accuracies (%) obtained by using the SVM classifier.

Dataset	Avg. Accuracy (%)
Blue whale (BW) + Noise	85.34
Fin whale (FW) + Noise	94.12
Blue whale (BW) + Fin whale (FW)	100

**Table 6 sensors-23-03048-t006:** The confusion matrix associated with the comparison method I for the ‘Blue whale (BW) + Noise’ dataset (the accuracy (%) is indicated in bold font and calculated as a ratio of the sum of diagonal values to the sum of all values × 100).

True class				
Predicted class		Blue whale (BW)	Noise	Specificity (%)
	Blue whale (BW)	89	40	69.0
	Noise	19	69	78.4
	Sensitivity (%)	82.4	63.3	72.8

**Table 7 sensors-23-03048-t007:** The confusion matrix associated with the comparison method I for the ‘Fin whale (FW) + Noise’ dataset (the accuracy (%) is indicated in bold font and calculated as a ratio of the sum of diagonal values to the sum of all values × 100).

True class				
Predicted class		Fin whale (FW)	Noise	Specificity (%)
	Fin whale (FW)	283	69	80.4
	Noise	71	292	80.4
	Sensitivity (%)	79.9	80.9	80.4

**Table 8 sensors-23-03048-t008:** The confusion matrix associated with the comparison method I for the ‘Blue whale (BW) + Fin whale (FW)’ dataset (the accuracy (%) is indicated in bold font and calculated as a ratio of the sum of diagonal values to the sum of all values × 100).

True class				
Predicted class		Blue whale (BW)	Fin whale (FW)	Specificity (%)
	Blue whale (BW)	51	34	60
	Fin whale (FW)	54	327	85.8
	Sensitivity (%)	48.6	90.6	81.1

**Table 9 sensors-23-03048-t009:** The average accuracies (%) obtained by comparison method II based on the Scattergram and CNN classifier.

Dataset	Avg. Accuracy (%)
Blue whale (BW) + Noise	72.26
Fin whale (FW) + Noise	81.64
Blue whale (BW) + Fin whale (FW)	48.20

## Data Availability

Not applicable.

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
