# Peer review of "A New Acoustical Autonomous Method for Identifying Endangered Whale Calls: A Case Study of Blue Whale and Fin Whale"

_sensors, 2023, doi:10.3390/s23063048_

Round 1

Reviewer 1 Report

In this paper, the author adopted deep learning classifier to automatically detect/classify the endangered whale calls. The whole looks interesting. Some suggestions are presented as follows.

1. From 2.2.2~2.2.3, the author are suggested to explain the main reason of the LSTM and other neural networks used in this paper. Some detail reasons are suggested to be provided.

2.  Figure 3 is suggested to be explained more.

3. Eq.14 should be discussed more. 

4. In Figure 4, what the relationship between CNN and LSTM used in this paper?

Author Response

Responses to Comments from Reviewer 1:

The author gratefully acknowledge the valuable comments by the Reviewer. We provide our actions to each comment. The reviewer comments are presented first and the corresponding replies are provided below each comment. Also, in the revised version, the revisions are highlighted by red text. We tried our best to carefully address every aspect you mentioned, and make revisions to improve the paper.

1.From 2.2.2~2.2.3, the author are suggested to explain the main reason of the LSTM and other neural networks used in this paper. Some detail reasons are suggested to be provided.

Re:  We have included the relevant information in section 1, pages 1-2 and 4th paragraph.

2. Figure 3 is suggested to be explained more.

Re: We have added the corresponding text for current Figure 4 in section 3.3, pages 7--8. 

3. Eq.14 should be discussed more.

Re: We have discussed it in section 3.5.1, page 9 and 1st paragraph.

4. In Figure 4, what the relationship between CNN and LSTM used in this paper?

Re: We have clarified it (current Figure 5) in section 4, page 11 and 4th paragraph.

We hope that the revised paper has been adequately addressed the comments and concerns of the reviewer.

Reviewer 2 Report

- The article's main concept(s)

In Europe, we have the EMSO-ERIC initiative that aims to explore the oceans and create a set of seafloor and water column observatories. In Canada, we have the Ocean Network Canada, in other countries we have different observatories and research cruise missions that gather acoustic data from the oceans. All this data can listen and detect seismic activity, detect marine life (whales, dolphins, sharks, …), and monitor all the acoustic noise on the ocean.

In this paper, the authors use datasets of acoustic data, with records of whales sounds, and tries to detect and classify endangered species such as Blue and Fin whales.

The author intends to implement Deep Learning techniques to detect and classify two sets of whales (Blue and Fin whales) and be able to understand and estimate the areas and the quantity of these endangered whales exists.

It is extremely important to know the quantity and the areas that should be protected to conserve these endangered marine species. Therefore, using datasets of whale calls and training models with sub-datasets with only blue and fin whale calls – would be able to automatically detect and classify which whale specie, its quantity, and areas of movement/breeding.

The author presents a deep learning LSTM classifier that uses the deep nonlinear features of wavelet scattering transform (WST) to automatically detect/classify the whale calls/sounds.

Using a small dataset from ONC acoustic datasets of whale calls, it was able to create a smaller dataset of 932 wav files (217 Blue, 715 Fin). Blue whale calls have a frequency of 20-100 Hz, while the Fin whale is around 5-100 Hz. The wavelet scattering transform would extract features, and then the LSTM would classify the information in different clusters. Long Short-term Memory is a recurrent neural network that is commonly used in deep learning developments/achievements.

The authors also compared its method with a relevant state-of-the-art Short-Term Fourier Transform technique, and against another state-of-the-art method that combines scattergram and deep convolutional neural networks.

- Overall Comment

In overall,

 This document shows a reasonable understanding of deep learning techniques and WST applied to acoustic signals topic and its key factors. The work has a good theoretical base, using useful references and information of general knowledge.

It presents a new development where the author uses commonly used techniques but applies them to whale calls to try to detect and classify two different endangered whale species.

It presents a slight novelty or technique, with the merge of WST and LSTM.

The conclusions/discussion resume the paper's contribution and explain common sense knowledge. On the other hand raises the importance of automatic acoustic sound identifiers techniques, as well as the importance of protection of majestic endangered species such as blue/fin whales.

Some regards/remarks:

The software is Audacity and not Audicity version 2.0.6

- Weak and Strong points

Strengths

v  Very good resume/comparison regarding deep learning techniques;

v  Wavelet scattering transform and Deep Learning;

v  Detect and classify whale calls – endangered species;

v  Well explained and good coherence in the results section;

Weakness

v  It would be great to have some software/toolbox available to the community;

v 

Author Response

Responses to Comments from Reviewer 2:

The author gratefully acknowledge the valuable comments by the Reviewer. We provide our actions to each comment. The reviewer comments are presented first and the corresponding replies are provided below each comment. Also, in the revised version, the revisions are highlighted by red text. We tried our best to carefully address every aspect you mentioned, and make revisions to improve the paper.

1. The software is Audacity and not Audicity version 2.0.6.

Re:  We have corrected it in section 2.1, page 3 and 2nd paragraph.

2. It would be great to have some software/toolbox available to the community.

Re: We have added a relevant remark in section 4, page 12 and last paragraph. We'll also upload the corresponding codes in the GitHub.

We hope that the revised paper has been adequately addressed the comments and concerns of the reviewer.

Reviewer 3 Report

The presented research aims at recognizing different species of whales by using a sound data processing algorithm based on wavelet analysis and neural network classification.  Overall, the paper is well written with a lot of details. 

The presented results outperform the previous (state-of-the-art) methods however, the complexity of the data processing is very high. By complexity I mean the number of parameters used along the data processing, i.e. resampling frequency, filtering, Hann window width, data sample length, or number of hidden layer neurons, firing functions, etc. 

It would make the significance of this research more impactful if the algorithm is less dependent on so many settings. I would also strongly recommend to graphically outline the entire signal processing process with intermediate data structures. Since the raw data are first preprocessed, the flow of the data manipulation has to be clearly identified. 

Author Response

Responses to Comments from Reviewer 3:

The author gratefully acknowledge the valuable comments by the Reviewer. We provide our actions to each comment. The reviewer comments are presented first and the corresponding replies are provided below each comment. Also, in the revised version, the revisions are highlighted by red text. We tried our best to carefully address every aspect you mentioned, and make revisions to improve the paper.

1. It would make the significance of this research more impactful if the algorithm is less dependent on so many settings.

Re:  We would address this issue in our future work by optimizing some parameters as mentioned in section 4, page 12, 6th paragraph and lines 305-307.

2. I would also strongly recommend to graphically outline the entire signal processing process with intermediate data structures. Since the raw data are first preprocessed, the flow of the data manipulation has to be clearly identified.

Re: We have added relevant information as well as Figure 1 in section 2.2.1, page 3. 

We hope that the revised paper has been adequately addressed the comments and concerns of the reviewer.

Round 2

Reviewer 1 Report

The author has revised the manuscript based on the reviewers' comments. I suggest to accept this manuscript.